# Melt-Spinnable Polyacrylonitrile—An Alternative Carbon Fiber Precursor

**DOI:** 10.3390/polym14235222

**Published:** 2022-11-30

**Authors:** Elena V. Chernikova, Natalia I. Osipova, Anna V. Plutalova, Roman V. Toms, Alexander Y. Gervald, Nickolay I. Prokopov, Valery G. Kulichikhin

**Affiliations:** 1Faculty of Chemistry, Lomonosov Moscow State University, Leninskie Gory 1, Bld. 3, 119991 Moscow, Russia; 2Topchiev Institute of Petrochemical Synthesis, Russian Academy of Sciences, Leninskii Pr. 29, 119991 Moscow, Russia; 3Institute of Fine Chemical Technologies Named by M.V. Lomonosov, MIREA—Russian Technological University, 119571 Moscow, Russia

**Keywords:** melt-spun fibers, polyacrylonitrile, cyclization of nitrile groups, thermo-oxidative stabilization, melt stability, extrusion, carbon fibers

## Abstract

The review summarizes recent advances in the production of carbon fiber precursors based on melt-spun acrylonitrile copolymers. Approaches to decrease the melting point of polyacrylonitrile and acrylonitrile copolymers are analyzed, including copolymerization with inert comonomers, plasticization by various solvents and additives, among them the eco-friendly ways to use the carbon dioxide and ionic liquids. The methods for preliminary modification of precursors that provides the thermal oxidative stabilization of the fibers without their melting and the reduction in the stabilization duration without the loss of the mechanical characteristics of the fibers are discussed. Special attention is paid to different ways of crosslinking by irradiation with different sources. Examples of the carbon fibers preparation from melt-processable acrylonitrile copolymers are considered in detail. A patent search was carried out and the information on the methods for producing carbon fibers from precursors based on melt-spun acrylonitrile copolymers are summarized.

## 1. Introduction

Synthetic and natural fibers are often produced by melt spinning by extrusion of the polymer melt through spinnerets with holes of a certain size, followed by jets stretching, cooling, and winding the solid fibers. During drawing, a slight decrease in the fiber diameter and an increase in its strength characteristics proceed [1,2]. Technical aspects of the melt spinning have been developed for numerous fiber-forming polymers and are summarized in one of the recent reviews [3]. The major economic advantage of the melt spinning procedure compared to solution spinning is the absence of solvent and coagulants, which eliminates the following steps in the fiber production: polymer dissolution, jets precipitation, fibers drying, solvent extraction and purification, and its recovery.

Some synthetic polymers (polyethylene, polypropylene, etc.), mesophase pitches, and biopolymers such as cellulose or lignin are used as carbon fiber precursors [4,5,6,7,8,9,10,11,12,13,14,15,16,17,18]. However, acrylonitrile (AN) copolymers are the most frequently used precursors for the production of carbon fibers [19,20,21,22,23,24]. Carbon fibers are widely used to prepare polymer composites [25] and their applications are being expanded intensively and are not limited now to aerospace and defense market, but include automotive, sporting goods, wind energy, compressed gas storage, etc. [26].

Their advantages include the ease of the synthesis based on radical polymerization in aprotic organic solvents (DMSO, DMF, etc.), concentrated water-salt media (NaSCN, ZnCl_2_), and water (emulsion, suspension, and precipitation polymerization) [27,28,29,30,31,32,33,34,35]; the variability of comonomers with different functionalities [36,37,38,39,40,41]; and controlling the molecular characteristics by changing the polymerization conditions [42,43,44,45].

Polyacrylonitrile (PAN) is a crystallizing polymer that is not typical for atactic macromolecules [46,47,48,49]. Its melting point is unexpectedly high (above 300 °C) and its direct determination by standard methods is complicated by the cyclization reaction that occurs at a lower temperature (approximately at 260–270 °C) than melting proceeds [50,51,52,53]. In its turn, cyclization leads to the change of the backbone structure due to formation of a ladder polymer (Figure 1).

The first attempts to find the melting point of PAN were conducted in the 1960s–1970s by dilatometry using PAN mixtures with solvents (DMF, γ-butyrolactone) [54] or by DTA of the AN copolymers with different comonomer (vinyl acetate) content [55]. As a result, it was estimated that melting point of PAN lies in the range of 317–322 °C and PAN has low values of the enthalpy (ΔH = 5 kJ⋅mol^−1^) and entropy of melting per repeating unit (ΔS = 8 J⋅mol^−1^⋅K^−1^). The influence of the solvents on the melting point of PAN is due to the dipole–dipole interactions which appear between nitrile groups and solvent molecules [56]. Based on these data and the X-ray diffraction results, the concept of the helical PAN conformation was formulated for the first time. It is the helical conformation that provides the formation of a long-range order during crystallization even in the absence of the configuration regularity in the chain structure. Later, the ultrafast DSC analysis at a scanning rate of 100 °C/min and higher gave rise to directly determine the melting point of PAN [57], the value of which coincided with the previously obtained values. 

The combination of two factors, namely the cyclization reaction proceeding below 300 °C and polymer melting above this temperature, makes it difficult to produce melt-spun PAN. As a consequence, the special approaches are required to reduce the melting point below the cyclization temperature, which are based on the decreasing of the interaction between nitrile groups of AN units. Therefore, when spinning PAN fibers, a dilemma arises: which method should be preferred, a traditional spinning from a solution or from a melt? The first method is performed under milder conditions, but it requires the solvent and non-solvent recovery and allows processing of relatively small amount of polymer, since its concentration in the spinning solution usually does not exceed 20–25 wt.%. The latter method requires a significant decrease of the melting point of the polymer, but it is free from the disadvantages of spinning from a solution.

The study of rheology of solutions and melts of AN copolymers and the processes of their spinning has been lasted for more than half a century. Significant progress in the production of high-strength and high-modulus carbon fibers has been achieved on the basis of the solution-spun PAN precursors [58,59,60,61]. In this case, a wide range of comonomers with different functionalities can be used. According to the classification proposed in the patent [59], these comonomers are able to increase (1) fiber density (vinyl monomers containing hydrophilic functional groups, such as carboxyl-, sulfonyl-, or amino-); (2) fiber formability, its orientation, and crystallization (alkyl (meth) acrylates, vinyl acetate); (3) stabilization rate (vinyl acids and amides); and (4) oxygen permeability (alkyl acrylates with a bulky substituent). The stress–strength properties of carbon fibers based on a melt-spun PAN precursor are far below [62,63,64]. Nevertheless, the melt-spinning retains its attractiveness due to the significant reduction in the cost of both textile and carbon fibers production.

Reducing the melting point of the polymer can be achieved in two ways: chemical and physical. In the first case, it is necessary to introduce a monomer into the PAN macromolecule, which will lower the melting point to an acceptable value. In the second case, a plasticizer should be used [65]. In practice, both approaches are used either separately or together. For textile fibers, no further heat treatment is required, while the production of carbon fiber demands to solve the next task: thermal oxidative stabilization (TOS) of precursor should proceed without its melting. Methods for production of melt-processable PAN are the same for textile fibers and carbon fiber precursors, while conducting of TOS requires a separate solution. Below, we will discuss the methods for production of melt-spun PAN and carbon fibers from them, described in the scientific and patent literature (see Appendix A).

## 2. Chemical Approach to Reduction of Melting Point: Selection of Copolymer Composition

The chemical approach to reduction of melting point of PAN is based on the introduction of comonomer units in the PAN macromolecule. These units will additionally disrupt the regularity of the chain and may lead to the transition of PAN to a melt state before its cyclization begins. The first information about the melting of AN copolymers is dealt with the vinyl acetate (VAc) as a comonomer [54]. An increase in its content in the copolymer from 2.7 to 38.5 wt.% lowers the melting point from 305 to 81 °C. In further studies [66], copolymers of AN of various compositions (from 2 to 40 mol.%) with vinylidene chloride (VDC), VAc, methyl acrylate (MA), ethyl vinyl ether (EVE), vinyl bromide (VB), and vinyl chloride (VC) were tested. These copolymers with close values of M_n_ ~55 kg⋅mol^−1^ and dispersity Đ = M_w_/M_n_ ~ 1.8 were synthesized by a radical polymerization at a fixed composition of the monomer feed throughout reaction. In terms of their ability to reduce the melting point of PAN, the comonomers are ranked in the descending order: VAc > EVE > VB > VDC > VC, which correlates with a decrease in the molar volume of the side substituent. For example, for AN and VAc copolymer, the melting point decreases from 176 to 140 °C with an increase in the VAc content from 1.6 to 11.0 wt.%; for VDC, the melting point decreases from 173 to 150 °C with an increase in the fraction of VDC from 14.2 to 36.3 wt.%. This effect is enhanced when passing from dry to wet copolymers. The trends of reduction of the melting point with an increase in the molar volume of the side substituent of comonomer and with a rise of its content observed for binary copolymers are applicable for both ternary copolymers, e.g., AN–VAc–VB, AN–VAc–VDC, AN–VB–VDC, AN–MA–VB, and AN–VB–VDC–VAc quaternary copolymer. 

### 2.1. Binary and Ternary Copolymers of Acrylonitrile with Methyl Acrylate

Thus, it is not difficult in principle to obtain melt-processable PAN. To do this, it is necessary to disrupt the sequence of AN units to a sufficient extent so that a helical conformation of the chain is not formed. For example, AN and VAc copolymers (10–25 mol.%) with M_η_ ~14–31 kg⋅mol^−1^ capable to melt were used to obtain microencapsulated phase change materials [67]. More attention is paid to MA, which has the similar ability to lower the melting point of PAN. In addition, MA and other alkyl acrylates have no accelerating or inhibiting effect on the rate of cyclization reaction [68,69]. However, they shift the onset of cyclization to higher temperatures and reduce the intensity of the exo-effect of cyclization and thermal-oxidative stabilization due to desequencing of AN units in the macromolecule [70]. Thus, MA additive in PAN separates the melting and cyclization processes and makes it possible to melt-spun a fiber. 

In the patent literature (see below), AN and MA copolymers capable of melt processing without cyclization have been described first in the mid-1990s. In the academic literature, they have been discussed in 2001–2003 in a series of articles [71,72,73]. The authors tried to find an analogue of the commercial sample supplied by Barex that contained 65 mol.% of AN, 25 mol.% of MA, and 10 mol.% of elastomers. It can form a stable melt at 180 °C at least for 30 min. Aiming for this, copolymers of AN and MA with the mole fraction of the latter from 2 to 15% have been synthesized in a DMF solution and in water (precipitation polymerization). The critical composition of the copolymer that provides its melt extrusion corresponds to 10 mol.% MA. Below this value, the copolymer cannot be processed from the melt due to the short melt lifetime even at low molecular weights (M_n_~20 kg⋅mol^−1^). Indeed, a few minutes after melting, a rise of viscosity starts due to the cyclization reaction. An increase in the molar part of MA in the copolymer from 10 to 15% leads to a decrease in the flow activation energy and an expansion of the range of MW of copolymers suitable for spinning fibers from the melt. The acceleration of viscosity moment is time-shifted with the decrease of MW of copolymer. So, AN copolymers with M_n_~20 kg⋅mol^−1^ and MA content of 10 and 15 mol.% withstand three successive melting-crystallization cycles withholding for 2 min at 200 or 250 °C without cyclization reaction [73]. These studies raised a number of questions. What are the roles of the monomer unit sequence, compositional inhomogeneity, and MWD of a copolymer in its flowability? 

A more detailed study was performed in [71]. However, it did not answer the questions posed, but only stated a number of positive findings. Comparative analysis of copolymers obtained by three synthesis routes-solution polymerization in DMF, precipitation, and suspension polymerizations in water showed that different compositions and MW of copolymers are required to form stable melts at 220 °C. The reason for this becomes clear if we take into account that the reactivity ratio of monomers varies for solution polymerization (r_AN_ = 1.22–1.29, r_MA_ = 0.61–0.96 [74]) and heterophase polymerization (r_AN_ = 0.83, r_MA_ = 1.17 [67]), as well as the dispersity of the resulting copolymers differs significantly. Unfortunately, the authors have varied all parameters simultaneously: copolymers composition, their compositional inhomogeneity, MW, and MWD. Nevertheless, an analysis of the results [72] allows one to conclude that the main factor affecting the viscosity of the melt is the MWD of the copolymer in addition to its composition. An increase in the proportion of the high molecular weight fraction leads to a noticeable increase in viscosity and a narrower temperature range of melt stability.

The melt-processable AN copolymers with 15 mol.% of MA and M_n_~20–40 kg⋅mol^−1^ obtained by precipitation and emulsion polymerization are described in [75,76,77]. These copolymers were used to produce hollow fibers with a tensile strength of 16 cN/dtex, elongation at break of 18.7%, and elastic modulus of 3 GPa [75]; films containing a microencapsulated phase change material [76] and membranes [77]. The formation of fibers, films, and membranes was conducted at 200–210 °C. 

The combination of the molding ability and an increase in the limiting oxygen index of the copolymer was achieved in [78] by using a phosphorus-containing comonomer-dimethylphosphonomethyl acrylate (5–7%). Its precipitation copolymerization with AN and terpolymerization with AN and MA have resulted in the binary and ternary copolymers with M_n_~35–55 kg⋅mol^−1^, Đ~5, and the limiting oxygen index 22–26 instead of 17–18, typical for PAN [79]. TOS of the copolymers proceeds in the temperature range of 239–255 °C. To decrease the melting point of copolymers, 22.5 wt.% of propylene carbonate is added. The latter is easily removed from the final fiber by washing with water. Finally, melt-spun fibers were produced at 175 °C with a tensile strength of 195 MPa, a Young’s modulus of 5.2 GPa, and an elongation at break of 19%.

However, the question remains: are these melt-spun copolymers suitable as carbon fiber precursors? To verify this, the rate of TOS of the spun fibers must be higher than that of the melting process. For copolymers with M_n_ = 110–130 kg⋅mol^−1^, Đ = 2.2–2.8, and MA content of 10–15 mol.%, melting is observed at 250 °C, but stability of the melt does not exceed 4 min. With an increase in temperature of 10 °C, the stability of the melt is lower, and cyclization is completed in about half an hour [80]. Melting of AN and MA copolymers of a lower MW (M_n_~20 kg⋅mol^−1^) proceeds at temperatures of 200–210 °C, while the TOS lasts for about a day at 220 °C [81]. To shorten the duration of TOS, an attempt was made to introduce a small amount of the third monomer (from 1 to 5 mol.%) into the copolymer. This monomer must be able to accelerate cyclization, and typically includes itaconic, methacrylic, or acrylic acids, as well as acrylamide. For terpolymers containing about 13 mol.% MA and no more than 3 mol.% itaconic acid or 4 mol.% of acrylamide, acrylic or methacrylic acids provide stable melts at 200 °C. An increase in the viscosity of terpolymer melts was observed at 220 °C. The rate of viscosity growth decreased in the following order: itaconic acid > acrylamide > methacrylic acid > acrylic acid.

A similar idea of using a comonomer facilitating cyclization was described in [80,82,83] as illustrated by ternary and binary copolymers of AN, MA, and dimethyl itaconate. Ternary copolymer with a content of AN of 80–90 mol.% forms a stable melt at temperatures of 175–210 °C, and the melt-spun white fiber is characterized by a tensile strength of 13.3–31.0 cN/tex and elongation at break of 10–30% [82]. A study of the thermal behavior of ternary copolymers with dimethyl itaconate content of 8.4–11.1 mol.% and MA of 1.8–3.0 mol.% and M_n_ = 65–110 kg⋅mol^−1^ and Đ = 1.7–3.0 showed that cyclization is fully completed in about 40 min at 230 °C and in 10 min at 260 °C [83]. Cyclization of binary copolymers with dimethyl itaconate content from 5 to 15 mol.% and M_n_ = 40–60 kg⋅mol^−1^ and Đ = 1.1–1.7 is completed in 3.5–4 min at 260 °C. The copolymer with the maximum content of dimethyl itaconate melts at 190 °C and copolymer with 10 mol.% of dimethylitaconate melts at 200 °C. The stability of the melt significantly decreased with a further increase in temperature due to the onset of the cyclization reaction [80]. The higher fusibility of the copolymers with dimethyl itaconate compared to the AN and MA copolymers can be explained by the large molar volume of the two side substituents of itaconate, which disrupts the regularity of the PAN chain. Despite the promise of using comonomers that shift the onset of cyclization to the region of low temperatures, there is practically no information on the mechanical characteristics of both melted PAN fibers and final carbon fibers.

### 2.2. Copolymers of Acrylonitrile with 1-Vinylimidazole

1-Vinylimidazole (VIM) is another potential monomer for obtaining the melt-processable PAN. VIM and poly (1-vinylimidazole) are water-soluble. Therefore, it is very difficult to synthesize copolymers of AN and VIM in water (emulsion, suspension, precipitation copolymerization). In this regard, copolymerization of AN and VIM is performed in organic solvents typical for PAN, e.g., DMF or DMSO [84,85,86,87]. A specific feature of the copolymerization of AN and VIM is the broadening of the MWD at high monomer conversions up to Đ > 2.5, which manifests itself in solution polymerization and is suppressed in the presence of a chain transfer agent. The information about the effect of VIM on the glass transition temperature is contradictory. Thus, its decrease is observed with an increase in the content of VIM in the copolymer [84], while the opposite observation is described in [87]. The molecular weight distribution plays a minor role in the thermal behavior of the copolymers of AN and VIM, while the copolymer composition plays an important role. The activation energy of the cyclization reaction grows linearly with an increase in the content of VIM in the copolymer. The higher the mole fraction of VIM in the copolymer, the slower the development of the ladder structure. In general, VIM can be considered as a monomer inert in cyclization. The rate of the formation of a ladder structure at a constant temperature differs sharply in argon atmosphere and in air. Thus, 10% conversion of nitrile groups into –C = N– groups is achieved after 2.5 h of heat treatment in argon at 225 °C or after 15 min in air at the same temperature [87]. This difference made it possible to find the optimal composition of the copolymer (18 mol.% VIM), at which the copolymer forms melt (190–200 °C) in an inert atmosphere before the onset of cyclization. After heating in air above 210 °C, the copolymers quickly lose their ability to dissolve and change color to dark brown due to the formation of a ladder structure. The melt-spun white fiber has a tensile strength of 1.4–1.6 cN/dtex and an elongation at break of 7.1–11.2% [84]. Further studies of the processes of TOS and carbonization of this precursor render it possible to produce carbon fiber. The tensile strength of the neat white fiber is 300–491 MPa, the stabilized fiber is 750–975 MPa, and the carbon fiber is 1.43 GPa; the elastic modulus of the initial fiber lies in the range of 90–110 GPa, the stabilized fiber is 128–158 GPa, and the carbon fiber is 156 GPa [85].

### 2.3. Copolymers of Acrylonitrile with Other Monomers

An increase in the thermal stability of PAN-based fibers was achieved by using styrene as a comonomer [88]. The copolymer obtained by suspension polymerization was characterized by M_n_ = 46 kg⋅mol^−1^ and Đ = 3.0, but its composition was not disclosed. In contrast to the monomers discussed above, styrene inhibits the cyclization reaction resulting in the shift of exo-effect to the high temperatures region and drastic reduction of its intensity [89,90]. The melting point of the copolymer is 231 °C, while cyclization starts above 290 °C and TOS starts above 260 °C.

The copolymerization of AN and methacrylonitrile is another way to produce melt-processable copolymer [91]. These copolymers with a mole fraction of AN from 20 to 95% are typically synthesized using a regulator of MW by emulsion radical copolymerization at a fixed composition of comonomers. Membranes molded from a melt of copolymers at 235 °C have high barrier properties.

Thus, the chemical approach makes it possible to obtain carbon fiber from melt-processable PAN, but its capabilities are most often limited either by a narrow temperature range of melt stability, or, conversely, by a large difference between melting point and cyclization temperature, which requires additional operations to prevent polymer melting at the TOS stage.

## 3. Physical Approach: Plasticization

An alternative solution of preparing the melt-spinnable PAN is plasticization. This technique is convenient both for the production of textile fibers and carbon fiber precursors. In the first case, it is easy to get rid of the correctly chosen plasticizer at the stage of fiber drying, which can provide an economic benefit compared to solution spinning. In the second, the removal of the plasticizer from the fiber will increase its melting point and the fiber will retain its shape during cyclization.

Two kinds of plasticizers have been analyzed: solvents in a limiting amounts and non-solvents playing a role of so-called structural plasticizers. Among solvents, the effect of lowering the melting temperature upon the addition of DMF or γ-butyrolactone to PAN was described in [54]. Later, it was found that the DSC method still makes it possible to determine the melting point of PAN and study the effect of solvents on its decrease [92,93,94]. 

### 3.1. Plasticization by Water (Non-Solvent)

As non-solvent, water deserves special attention being simultaneously precipitant for PAN and plasticizer, which has the property of lowering the melting point of the AN homopolymer and its copolymer with VAc [92,93]. This is clearly illustrated in [92] using the example of homo-PAN and two copolymers of AN. The authors found that in the absence of water at high scanning rates (80 and 160 °C/min), the DSC curves of PAN in inert atmosphere show a weak endothermic melting peak with a maximum at a temperature of 345–360 °C followed by an intense exothermic peak corresponding to cyclization. For the copolymer of AN and acrylic acid (3.3 mol.%), the exotherm shifts to the region of low temperatures and the melting peak is not observed under any conditions. On the contrary, for the terpolymer of AN, VAc (7 mol.%), and sodium metallylsulfonate (0.3 mol.%), the exothermal peak shifts to the region of higher temperatures, and melting occurs at a temperature of 307 °C. In the presence of an equal amount of water relative to the polymer, an intense melting peak for binary copolymer is observed at 184 °C; for the terpolymer, melting is detected at 157 °C, but for the copolymer of AN and acrylic acid, melting still cannot be recorded. 

A more detailed study of the effect of the amount of added water on the melting point of AN copolymers is described in [93,94]. A decrease in the melting point of PAN from 320 to 185 °C is observed with increasing water content from 0 to 35 wt.%. Moreover, a further increase in the water content has no effect on the melting point. Water has the same effect on AN copolymers. For example, the melting point of the copolymer AN and VAc (~2 mol.%) is ~285 °C in the absence of water, and it decreases to ~165 °C when 67 wt.% of water is added [93]. The authors have determined the dependence of the melting point of AN and VAc copolymers on the mole fraction of VAc in the copolymer and showed that it is possible to estimate the melting points T;_melt(PAN)_ of “dry” (1) and T;_melt(PAN/water)_ of “wet” (2) copolymers of arbitrary composition (*x_i_*) based on the model of a semi-crystalline polymer containing defects, e.g., comonomer units:(1)$$\frac{1}{T_{m.p(\mathrm{PAN})}}=\frac{1}{T_{m.p,0(\mathrm{PAN})}}+\sum_{i=1}^{n-1}k_{i}x_{i}$$
(2)$$\frac{1}{T_{m.p(\mathrm{PAN}/\mathrm{water})}}=\frac{1}{T_{m.p,0(\mathrm{PAN}/\mathrm{water})}}+\sum_{i=1}^{n-1}k_{i}x_{i}$$
where *T;*_m.p,0(PAN)_ = 617 K, *T*_m.p,0(PAN/water)_ = 456.8 K, *x_i_* is the molar fraction of monomer *i*, *n* is the number of comonomers (2 is a binary copolymer, 3 is a terpolymer, etc.), and *k_i_* is the melting point lowering constant. The latter is 3.37 × 10^−3^ K^−1^ for MA and 3.34 × 10^−3^ K^−1^ for VAc [90]. The reciprocal melting points of the wet and dry copolymers differ by the quantity ($1/T_{m.p,0\left(\mathrm{PAN}\right)}-1/T_{m.p,0\left(\mathrm{PAN}/\mathrm{water}\right)}$). Therefore, the dry polymer melting point can be calculated from the wet polymer value, and vice versa. This relationship is useful when thermal degradation prevents the direct determination of the dry polymer melting point.

These fundamental studies led to the idea of using various solvents to produce melt-processable PAN and its copolymers. The plasticization of PAN with water to produce a melt-spun polymer was first described in the late 1970s [95]. Water is able to lower the melting point and melt viscosity, and also suppresses cyclization reaction by blocking nitrile groups [93,94]. The amount of water is a crucial factor in the processing of melt-spun PAN. It is believed that water is capable of hydrating nitrile groups and is completely miscible with PAN. At the same time, free water, i.e., water, which remains after full hydration of the nitrile groups, is separated into a single phase. 

The study of the interaction of water with AN copolymers and the search for its optimal content in melt-processable PAN is the subject of a large number of publications since the 1970s and continuing nowadays. Thus, it was shown that the regions of complete mixing of PAN with water and the phase of PAN depend on the temperature and water content in the system, which is clearly reflected in the composition diagram (Figure 1) [96]. Area 1 corresponds to a single liquid phase, in which the molten PAN is completely miscible with water, forming some kind of a complex or alloy. Area 2 corresponds to a two-phase system in which the melt (PAN–water “complex”) and free water coexist. Area 3 accords with a single phase of the mixture of PAN–water “complex” that did not turn into the melt, while area 4 corresponds to a two-phase region consisting of the “hard alloy” of PAN with water and free water. Phase diagrams for AN copolymers look similar, except that the molten copolymer is completely miscible with water at other values of temperature and water content [97].

PAN (M_η_ = 60 kg⋅mol^−1^) suspension containing 81 wt.% of water is easily extruded at 150 °C and high pressure (123 MPa) [98]. The water content can be reduced to 30 wt.%. However, in the absence of water, PAN is incapable of melting even at higher pressures. Copolymers containing at least 85% of AN and plasticized with water or glycerol, alcohols, ethers, etc. can be also subjected to extrusion. The color of the fiber changes from white to yellowish during extrusion, indicating partial cyclization of the nitrile units. Later, the fiber was produced at 202 °C from the mixture of 25 wt. parts of copolymer AN and Vac (7 mol.%) with M_η_ = 110 kg⋅mol^−1^ and 0.5 wt. parts of Na-carboxymethylcellulose plasticized with 300 wt. parts of water (region 2 in Figure 1) [99].

The thermal stability of a hydrated melt of a commercial copolymer of AN and Vac (12 wt.%) from Tae Kwang Industrial Co (Korea) was studied in [100,101,102,103]. The melting point of the copolymer with M_η_ = 88 kg⋅mol^−1^ in the absence of water is equal to 270 °C and it falls to 156 °C at ~23 wt.% of water whereafter no longer changes. When annealing the melt of the copolymer containing 23 wt.% of water at 160–180 °C, its melting point rises with increasing annealing time and the faster, the higher annealing temperature. Meanwhile, the intrinsic viscosity of the copolymers in DMF, whose melts containing water were subjected to annealing at temperatures of 160–180 °C during different times, decreases with time of annealing, regardless of annealing temperature, indicating the absence of cross-linking [100]. 

The behavior of hydrated melts of PAN and AN copolymers with MA, whose mole fraction was varied from 3.6 to 16.5%, is described in [104]. The melting point of polymers containing 50 wt.% of water decreases from 184 °C (PAN) to 133 °C (copolymer with 10.2 mol.% MA).

An additional lowering of the melting point can be achieved by introducing a small amount of co-solvent-ethylene carbonate [101] or DMF [102] to non-solvent. However, if a hydrophilic polymer (e.g., polyacrylic acid, polyvinyl alcohol, or polyethylene oxide) is added to the wet AN copolymer, the effect of lowering the melting point of copolymer is reduced by partially losing water to hydrate the hydrophilic polymers. 

Obviously, the water content in the melt can be varied over a wide range, but it is necessary to determine its optimal amount. This can be performed empirically, or one can first determine the amount of water that hydrates the nitrile groups of the polymer to a given degree at a selected temperature. This technique based on the combination of DTA and Raman spectroscopy was described in a patent and applied to producing fibers from PAN its binary and ternary copolymers [105]. For example, the spinning of the terpolymer AN (87.6–93.6 mol.%), MA (6–12 mol.%), and sodium styrenesulfonate (0.4 mol.%) from the melt was conducted at 171–180 °C, pressure of ~6.9 MPa, and a mass ratio of polymer and water equal to 100:25. As a result, a fiber with a core–shell structure containing a large number of voids and longitudinal grooves was obtained from melt-spun PAN. It is possible to improve the fiber morphology by adding a co-solvent to the copolymer, such as DMF, propylene carbonate, acetonitrile, etc. [106]. Another example includes the melting of copolymer of AN and VAc (12 wt.%) with 23 wt.% of water and additionally 5 wt.% of ethylene carbonate which was spun at 150 °C [101]. In this case, fibers were obtained from the melt in the absence and in the presence of ethylene carbonate and are characterized by a tensile strength of 315 and 360 MPa, an elastic modulus of 3.9, and 4.9 GPa, respectively. The fiber density was 1.15 g/cm^3^, which is less but comparable to the density of acrylic fibers obtained by solution spinning. A lower density indicates the porous structure of the resulting fibers. The formation of plexifilaments consisting of irregularly shaped fibers intertwined with each other and forming a network by extrusion of a dispersion of binary copolymers of AN (25–45 wt.%) and water (region 2 in Figure 2) at pressure of 3.4–10.3 MPa and high temperatures (240–290 °C) resulted in rapid removal of water as described in [107,108]. 

The preparation of the fibers with a core–shell structure from the melted PAN plasticized with water is described in [109]. In this case, PAN was partially cyclized during extrusion. Thus, a common undesirable phenomenon is observed in the mentioned research: the fiber formation is accomplished with undesirable uncontrolled diffusion of water. Therefore, strict control of the water evaporation rate, temperature, and fiber shrinkage is necessary to prevent the formation of a core–shell morphology and reduce the number of possible defects. This problem was partially solved in [95,110,111,112,113,114,115,116,117], where the fiber spinning process was performed as follows. Controlled evaporation of water from the extrudate is carried out by maintaining the required level of humidity in the fiber spinning zone. The temperature is chosen in the range between the minimum melting point (intersection of curves in Figure 1) and the glass transition temperature of the polymer in order to provide the required fiber drawing ratio [97]. Fiber drawing, which leads to reduction of the number of formed defects, is conducted via several stages, gradually increasing drawing multiplicity to ~25 and higher [97,112]. After that, highly oriented fibers were dried to remove water completely [113].

It is obvious that the removal of the plasticizer (water) will lead to the increase of the melting point of the fiber. Therefore, these fibers should be subjected to TOS and carbonization without fiber melting that is necessary to produce the carbon fiber. For the first time, carbon fibers obtained from a precursor based on plasticized melt-spun PAN were described in [97]. The chemical composition of the copolymers was not disclosed; the available information was limited to the mass fraction of the comonomer in the polymer samples as well. All copolymers are characterized by M_η_ ranging from 0.7 to 2.5 kg⋅mol^−1^. The draw ratio of the melt-spun white fiber varied from ~2 (5 samples) to 19 (1 sample). All fibers were characterized by a strength of ~3 MPa and a Young’s modulus of about 100 GPa. Fibers based on the copolymer with a comonomer content of 7 wt.% passed through three-stage stabilization at temperatures of 220, 250, and 270 °C, and fibers produced from the copolymer containing 3 wt.% of comonomer at 260, 280, and 290 °C. The choice of the TOS conditions allows assuming that the comonomer accelerates PAN cyclization, i.e., contains carboxylic or amide functionalities. The stabilized fibers had a heterogeneous structure. Nevertheless, after carbonization at 1200 °C, the mechanical characteristics of the carbon fibers were acceptable: a strength of 2.5 GPa and an elastic modulus of 170–215 GPa. Thus, melt-processable PAN containing water as a plasticizer is promising as a carbon fiber precursor.

### 3.2. Plasticization by Water-Organic Additive Mixtures 

Despite the fact that the use of water as a plasticizer is economically and ecologically attractive, organic species or their mixtures with water as plasticizers have also been considered in recent years. For example, five co-plasticizers, namely 2-ethyl-2-oxazaline, N-methylpyrrolidone, DMF, adiponitrile, and acetonitrile, were compared in terms of their ability to lower the melting temperature of AN and MA copolymer (4.4 mol.%) with M_n_ = 123 kg⋅mol^−1^ and Đ = 1.9 [118]. It turned out that the melting point of a hydrated copolymer lowers in the following order of the co-plasticizers: N-methylpyrrolidone ≈ DMF < adiponitrile < acetonitrile, while 2-ethyl-2-oxazaline, on the contrary, increases the melting point of the copolymer. This different behavior is caused by the different action of the listed compounds on the copolymer-water system. However, acetonitrile swaps with adiponitrile in terms of their ability to maintain melt stability (Figure 2). Melting points of mixtures copolymer/co-plasticizer/water are equal to 136 °C for adiponitrile/water = 15/15 wt.% (1), 134 and 142 °C for acetonitrile/water = 15/15 (2), and 10/20 wt.% (3), respectively. According to DSC data, the stability of the melts at 180 and 190 °C is kept for about 4 h; after that, it sharply decreases. This effect manifests stronger in the case of acetonitrile. The dependencies of viscosity on time at a fixed melt temperature and a shear rate of 161.3 s^−1^ have the same trend but show significantly shorter times due to the difference in the experimental conditions (pressure, shear rate). 

The direction with co-plasticizers was continued in [119,120,121] using the same copolymer. In a more detailed study of the copolymer/acetonitrile/water system, in which the ratio of acetonitrile and water was changed, the authors came to the conclusion that the copolymer/water system (30 wt.%) is the best option: a minimum melt viscosity is observed, and the melt has high stability at 180 °C [119]. It is somewhat surprising because under isothermal conditions, the cyclization of the copolymer of AN and MA occurs already at 180 °C, which suggests that the copolymer obtained by suspension polymerization could contain a residual initiator, which initiates cyclization at a sufficiently lower temperature. If ethanol, which is a precipitant for PAN, is used instead of acetonitrile, a similar tendency is observed [120]. Moreover, the fibers formed in the absence of alcohol or acetonitrile are more defective due to the formation of micropores because of difficult control over the rate of water diffusion during fiber drying.

### 3.3. Plasticization by Organic Solvents and Non-Solvents

Another popular plasticizer for melt-spun PAN is glycerol. The paper [122] describes a copolymer of AN and VAc (6 wt.%) with M_n_ = 138 kg⋅mol^−1^ plasticized with glycerol. The introduction of 30 wt.% of glycerol lowered the melting point of the copolymer from 290 to 217 °C, i.e., its efficiency is significantly lower in comparison with water. Nevertheless, the research was continued with a similar copolymer of lower MW (M_n_ = 45 kg⋅mol^−1^) [123]. The process of obtaining melt-processable PAN turned out to be more complicated, since the authors first mixed the copolymer with 28 wt.% of glycerol and 2 wt.% of additives described in the patent [124], subjected it to extrusion at 200 and 220 °C, and crushed the prepared fibers. At the second stage, the fibers from plasticized copolymer entered the second extruder, in which the main process of melting the plasticized copolymer and fiber spinning took place. However, the fiber still had a porous structure. According to IR spectroscopy data, TOS processes in the fiber started above 240 °C. Unfortunately, the authors do not provide data on the fiber melting point and the presence/absence of residual glycerol. Therefore, there are no grounds yet to conclude that it is promising for obtaining melt-spun PAN. 

The other attempt to use glycerol was made in [125] for a copolymer of AN with 1-vinylimidazole (21 mol.%) and a tetrapolymer of AN, VIM (2.5 mol.%), MA (14.9 mol.%), and acrylic acid (3.1 mol.%) with MW in the range 30–60 kg⋅mol^−1^. The spinning was conducted at 180 °C and then the fiber was additionally drawn at 150 °C. In the absence of glycerol, increasing capillary diameter and decreasing shear rate and capillary length to diameter ratio increased extrudate stability. The following parameters turned out to be optimal: the capillary diameter is 1.2 mm, its length is 6 mm, and the shear rate is 35.4 s^−1^. However, the resulting fibers turned out to be brittle, and the ultimate strength values were 50 and 35 MPa for binary copolymer and tetrapolymer, respectively. The introduction of 15 wt.% of glycerol lowered the viscosity of the melt significantly and increased its stability. The fibers had a porous structure and contained residual glycerol, which affected their mechanical properties. The tensile strength before drawing was 230 and 260 MPa for the copolymer and tetrapolymer, and after drawing 379 and 369 MPa, the value of Young’s modulus was not given.

Propylene carbonate can be used also as a plasticizer [126]. PAN becomes soluble in propylene carbonate at 130–150 °C. At a mass ratio of PAN (intrinsic viscosity in DMF is equal to 1.5 dL/g) and propylene carbonate of 50:50, the melt retains its stability up to 220 °C, which makes it possible to extrude. In this case, there is no need to maintain high pressure as opposed to the use of water. However, propylene carbonate having a high boiling point remains in the fiber after cooling, and the fiber requires additional purification. Recently, propylene carbonate was used to plasticize AN and MA copolymers obtained by precipitation polymerization [127]. The authors varied several parameters at once: the composition of the copolymer (6.4–13.3 mol.% MA), M_n_ from 19 to 47 kg⋅mol^−1^, and dispersity Đ from 2.1 to 4.0. As a result, the best behavior was shown by a copolymer containing 8.1 mol.% MA with M_n_ = 34 kg⋅mol^−1^, which melted in the presence 22 wt.% of propylene carbonate and was extruded at 175 °C. Since the plasticizer is not volatile, a method was developed to remove it from the fiber by blowing with hot air, then immersing in a hot water bath (90 °C) for 5 min, additional washing, and drying. By washing and stretching the fiber, it was possible to reduce its porosity. The tensile strength of the fiber not washed from the plasticizer is 280 MPa, the Young’s modulus is 7.9 GPa, and for the washed fiber, they are 370 MPa and 9.1 GPa.

### 3.4. Eco-Friendly Plasticization

In parallel, various scientific groups have proposed alternative environmentally friendly options for replacing water as a plasticizer on CO_2_ and ionic liquids. Let us compare them in terms of their ability to lower the melting point of AN copolymers with the plasticizers discussed above.

#### 3.4.1. The Use of CO_2_ as a Plasticizer of AN Copolymers

It is known that CO_2_ can be absorbed by amorphous and to a lesser extent crystallizing polymer, and as a result, their glass transition temperature decreases [128,129,130,131,132,133]. An increase in the polarity of the polymer drives up the increase of the amount of CO_2_ dissolving in polymer [128]. The AN copolymer from Barex, containing 25 mol.% MA and 10 mol.% elastomer, was used to study the effect of CO_2_ on the melt viscosity [134]. Obviously, the choice is due to the fact that this copolymer is amorphous, and absorption of CO_2_ will be easier than in the case of the semi-crystalline copolymers with a high content of AN. In addition, the high content of MA ensures its thermal stability at 200–220 °C. The authors showed that at 120 °C and elevated pressures of 10.3 and 17.2 MPa, the copolymer is saturated with CO_2_ within 6 h. The higher the pressure, the higher the CO_2_ content in the polymer. The maximum amount of absorbed CO_2_ is 6.7 wt.%, which leads to a decrease in the glass transition temperature of the copolymer by 31°C. It is important to note that in this case, it is possible to reduce the melt viscosity. Thus, the viscosity of a copolymer melt saturated with CO_2_ at 17.2 MPa at 180 °C corresponds to the melt viscosity of a copolymer not containing CO_2_ at 210 °C. The melt-spun fiber obtained from the copolymer with absorbed CO_2_ is characterized by low porosity.

In practice, AN copolymers can be used as carbon fiber precursors, if the comonomer content is ~15 mol.% or less. An increase in the proportion of AN in the copolymer with MA predictably increases the degree of crystallinity of the polymer and leads to a decrease in the amount of absorbed CO_2_. There are 5.6 and 3.0 wt.% of absorbed CO_2_ at the mole fraction of AN in the copolymer 85 and 90% [135]. For these copolymers, the glass transition temperature decreases by 37 and 27 °C, and the melt viscosity by 61 and 56%, respectively. As a result, the melt-processing temperature of the copolymer can be reduced by 26 and 9 °C at AN content in the copolymer of 85 and 90 mol.%. Obviously, with a further increase in the proportion of AN in the copolymer, the absorption of CO_2_ becomes insignificant, and this approach becomes unsuitable for production of melt-spun PAN.

From a practical point of view, the method of polymer plasticizing with CO_2_ is inconvenient. This problem was attempted to be solved by melting the polymer first and then adding CO_2_ flow to the melt at a given rate, followed by raising the pressure in the system and extruding the melt with absorbed CO_2_ [136]. The authors showed that a copolymer with MA content of 15 mol.% can be spun via the melt at 194 °C using plasticization with CO_2_. To reduce porosity, the extrudate must be subjected to additional pressure. Unfortunately, there are no literature data on the mechanical characteristics of the fibers, and it is not possible to estimate the residual amount of CO_2_ in the fiber.

#### 3.4.2. The Use of Ionic Liquids as Plasticizers of AN Copolymers

Another approach that can be considered as environmentally friendly is based on the application of ionic liquids, which are attractive due to their high solvency, non-volatility, and ease of regeneration. Interestingly, ionic liquids themselves, e.g., dialkylimidazolium salts (Figure 3), may serve as precursors for producing micro- and mesoporous carbon. Thus, it was shown that dialkylimidazolium salts both themselves and encapsulated in a silicone matrix form highly porous N-doped carbon with a high yield under thermolysis in a nitrogen atmosphere when heated from room temperature to 800 °C at a rate of 10 °C/min and kept for 1 h at 800 °C [137,138].

PAN and its copolymers are highly soluble in the same ionic liquids. For example, 1-butyl-3-methylimidazolium chloride is a good solvent for copolymers of AN with MMA (10 wt.%) with M_η_ = 45–60 kg⋅mol^−1^ and itaconic acid (96:4) with M_w_ = 79 kg⋅mol^−1^ and Đ = 1.8 [139,140]. In the range of dilute and semi-dilute solutions, the behavior of PAN (MW 75 kg⋅mol^−1^) in 1-butyl-3-methylimidazolium bromide is similar to the behavior of polymer solutions in traditional organic solvents, such as DMF [141]. Concentrated solutions of copolymers of AN with MMA and AN with itaconic acid in 1-butyl-3-methylimidazolium chloride (16–24 wt.%) also behave in a typical way. Their viscosity increases, and the Newtonian flow region decreases with increasing both polymer molecular weight and its concentration and decreasing the temperature. At elevated temperatures (50–100 °C), the loss modulus is higher than the storage modulus, and at lower temperatures (less than 40 °C), vice versa. When cooled, concentrated solutions turn into a gel, and when heated, they return to their original state, demonstrating the reversibility of the gel–solution transition.

1-Butyl-3-methylimidazolium chloride is able to plasticize AN copolymers, e.g., AN/MA/acrylamide (97.5/2.2/0.3%) terpolymer with M_η_ = 150 kg⋅mol^−1^ [142]. The addition of 15 wt.% of ionic liquid lowered the glass transition temperature of the terpolymer to 34 °C, and the melting point to 177 °C. The authors believe that imidazolium salt is capable of forming hydrogen bonds with nitrile groups according to Figure 2, and thus reduces the interaction of nitrile groups within the chain.

Investigation of the dependence of melt viscosity on time for terpolymers containing 20–30 wt.% of ionic liquid in the range of temperatures 180–200 °C showed that the melt stability is low. It is kept for 10 min for all samples at 180 °C, while the same time is held at 190 °C only at imidazolium salt content of 30 wt.%; at 200 °C, the viscosity starts to grow slightly after 4 min even at a high proportion of ionic liquid (30 wt.%). However, the authors have succeeded in performing melt-spinning of terpolymer. During molding, partial cyclization of the nitrile units occurred, which was confirmed by IR spectroscopy. DSC analysis of the formed fiber showed the presence of one exothermic peak with a temperature of cyclization onset of 210 °C and a maximum of 273 °C. The authors believe that a terpolymer plasticized with imidazolium salt can be used as a carbon fiber precursor.

A commercial PAN sample from Shanghai Jinshan Petrochemical Company with M_η_ = 78 kg⋅mol^−1^ was plasticized with the same ionic liquid in a mass ratio of 40:60 and extruded at 180–210 °C [143,144]. The authors showed that the DSC thermograms of the polymer after 14 min of extrusion shift to the region of high temperatures with an increase in the processing temperature. This is due to the occurrence of partial cyclization during extrusion, which agrees with the data of [142]. The degree of cyclization increases from 6 to 24% with an increase in the extrusion temperature from 180 to 210 °C. The same PAN and 1-butyl-3-methylimidazolium chloride were used in [145] to produce fibers by melt spinning at 180–200 °C. The obtained fiber was stretched in two coagulation baths with water, which, obviously, made it possible to wash PAN from the imidazolium salt, which is highly soluble in water. It is important to note that the fiber has a smooth surface, and the degree of crystallinity and dimensions of PAN crystallites in the fiber are comparable to commercial samples obtained by solution spinning. In this case, TOS of melt-spun PAN begins at a lower temperature and proceeds over a wider temperature range than in the case of fibers formed from a solution. Of greater interest are the mechanical properties of a fiber with 15 µm in diameter and linear density of 1.8 dtex. The tensile strength is 7 cN/dtex and the elongation at break is 9.3%. Thus, the structural and mechanical characteristics of the fiber produced from melt-spun PAN plasticized with an ionic liquid show the promise of this method for the production of carbon fiber.

The effect of the nature of ionic liquids based on imidazolium salts (Figure 4) on the plasticization of an AN–VAc copolymer (7 mol.%) with MW 160 kg⋅mol^−1^ and study of its processing through a melt is described in [146]. The polymer and imidazolium salt were taken in a weight ratio of 3:7.

Increasing the size of the substituent or replacing the chloride anion by the bromide anion leads to decrease in the plasticizing effect of the ionic liquid. Thus, the glass transition temperature of PAN is in the range from 72 to 86 °C, and the melting point is in the range from 108 to 157 °C, depending on the nature of substituent and anion. Spinning was carried out from a homogeneous melt at 150 °C. The authors have compared the structure and properties of the spun fiber and the fiber purified from the residues of ionic liquid by washing with water. The fiber diameter decreases after the removal of the ionic liquid, while the diameter dispersion increases. At the same time, the porosity of the fibers remains low; their structure is smooth and fairly uniform. It follows from the DSC data that partial cyclization occurred during spinning. It is interesting that after the removal of ionic liquid residues, the exotherm peak shifts to low temperatures, which is favorable for shortening the TOC time. The mechanical characteristics of the fibers are improved twice or more after washing the fibers with water. After purification, the elastic modulus is 5–6 GPa, the tensile strength is 90–120 MPa, and the elongation at break decreases to 20–45%.

The search for ionic liquids for melt production of PAN fibers was continued in [147] using 1-ethyl-3-methylimidazolium salts with various cations: chloride, bromide, dicyanamide, and tricyanamide. In this case, PAN with M_n_ = 52 kg⋅mol^−1^ and Đ = 2.5 is used and the weight ratio of PAN and ionic liquid is equal to 3:7. Molding was conducted at 160 °C. The authors have compared the performance of the fibers before and after washing with water. In the first case, ionic liquids are arranged in the following order according to the decrease in the elastic modulus of the fibers: chloride > bromide > dicyanamide > tricyanamide. Thus, for a spun fiber prepared in the presence of 1-ethyl-3-methylimidazolium chloride, the elastic modulus is 4.4 GPa and the tensile strength is 60 MPa. The elastic modulus is doubled after purification and is equal to 8.9 GPa. The elongation at break of the fibers increases from 75 to 300% in the series of anions: bromide < chloride < dicyanamide < tricyanamide before washing. Naturally, it falls after purification, e.g., in the case of tricyanamide it reduced from 209 to 22%, while in the case of chloride it decreases from 81 to 49%.

On the whole, the works mentioned above demonstrate the promise of using ionic liquids to produce fibers with high mechanical performance from melt-spun copolymers. The only drawback is the necessity of an additional stage, i.e., washing the fibers from the residual ionic liquid and its further regeneration.

An ionic liquid such as 1-butyl-3-methylimidazolium chloride can also be used as a polymerization medium. This approach is described in [148], where an AN–MA copolymer (3.7–7.6%) was obtained by polymerization of a 20% solution of monomers in an ionic liquid under the action of AIBN at 65 °C. Unfortunately, spinning procedure was conducted out from a 12% solution of copolymers in an ionic liquid at 90 °C using a coagulation bath with cold water (10 °C), but not from a melt. However, fibers with a core–shell structure were obtained with tensile strength equal to 2.0–4.7 cN/dtex depending on the drawing degree.

One way to reduce the cost of carbon fiber production is to mix the AN copolymer with a cheaper component, which can be, for example, lignin [149]. Lignin (M_w_ = 285 kg⋅mol^−1^, Đ = 2.2) isolated from poplar shavings was subjected to esterification with butyric anhydride to reduce hydrophilicity. After that, it was added to the copolymer AN with methacrylic acid (5.8 wt.%) with M_w_ = 85 kg⋅mol^−1^ manufactured by “Good Fellow” in a weight ratio of 10–30%. 1-Butyl-3-methylimidazolium chloride taken in the equivalent ratio with respect to the AN copolymer was used as a plasticizer. The introduction of an ionic liquid leads to a decrease in the melting point of AN copolymer, while the presence of lignin does not affect the thermal properties in any way, resulting on the whole in the value of melting point about ~120 °C. However, if the ionic liquid is replaced by DMF, then the PAN melting point increases with an increase in the lignin content [150]. The viscosity of the melt and the limit of plasticity increase in the temperature range of 150–200 °C with an increase in the proportion of lignin in the mixture of the copolymer and ionic liquid. Interestingly, the value of the plasticity limit is practically independent on temperature at a lignin content of up to 20 wt.%, but it decreases by more than two times with an increase in temperature by 50 °C at 30 wt.% of lignin. As a result, the optimum melt spinning temperature depends on the concentration of lignin in the mixture. Unfortunately, there is no data on whether the partial cyclization of nitrile units during the molding process proceeds as described above. There is also no information on the mechanical properties of the fibers. Therefore, it is too early to draw conclusions about the rationality of using this approach to produce cheap precursors of carbon fibers by melting.

## 4. How to Conduct Stabilization of the Fiber without Its Melting? Problem and Its Solution

As it was mentioned above, stabilization of the just prepared melt-spun PAN-based fiber may result in the fiber melting. To hold the fiber shape during stabilization, several approaches can be used that are discussed below.

### 4.1. Chemical Modification at the Synthesis Stage

Chemical modification of PAN by introducing a comonomer as an internal plasticizer makes it possible to decrease the melting point of the copolymer and makes it lower than the temperature of cyclization reaction of nitrile groups. However, in this case, there is a problem with the implementation of TOS. If the polymer is melted at a temperature before the start of cyclization, then the fiber will lose its shape completely or partially turn into a melt during TOS. As mentioned above, attempts were made to solve this problem by introducing a small amount of a third comonomer containing an acid group into PAN at the synthesis stage [84,85,86,87]. As a result, TOS in the spun fiber proceeds at a high speed although it starts above the melting point. Unfortunately, this process was not completed to the end, i.e., to the carbon fiber stage, and this is probably not the optimal solution.

Another version is based on the difference in the thermal behavior of the polymer in an inert and air atmosphere, as is the case of the copolymer of AN and 1-vinylimidazole [88,89,90,91]. At a certain composition and MW of the copolymer, the cyclization of the copolymer occurs at a temperature above the melting point, while TOS occurs below the melting point. The first carbon fiber described in the literature made from a melt-spun AN–VIM copolymer can be classified as medium-strength: a tensile strength is equal to 1.43 GPa and an elastic modulus of 156 GPa [89].

The copolymers of AN and N,N’-substituted acryloamidines (Figure 5) have the same property [151].

N,N’-substituted acryloamidines do not affect the glass transition temperature of PAN: it is 97 ± 1 °C at a comonomer mole fraction of 1–19%. At the same time, they lower the melting point of the polymer, which, for example, for a copolymer with an acryloamidine mole fraction of 18–19%, R = R’ = 2-propyl, and M_n_ = 20–24 kg⋅mol^−1^ and Đ = 1.8–2.0 is ~140 °C. In an air atmosphere, TOS starts at ~140 °C and proceeds very intensively, while in an inert atmosphere, the intensity of the heat flux during cyclization is 5–6 times lower, and its onset is observed at temperatures 10–15 °C higher. An additional decrease in the melting point to ~130 °C is achieved by the addition of 10% DMSO or DMF. As a result, fiber is formed by extrusion of the melt in an inert atmosphere, and TOS is realized by heat treatment in air at 250 °C. The stabilized fiber was subjected to carbonization at 1400 °C in an inert atmosphere and carbon fiber was obtained with the following characteristics: tensile strength is equal to 0.94 GPa and elastic modulus to 85 GPa. It should be noted that both the precursor fiber obtained using a comonomer with R = R’ = 2-propyl and carbon fiber had a smooth surface, and no porous structure was observed. Replacing one isopropyl group with a *tert*-butyl group (R = *tert*-butyl, R’ = 2-propyl) leads to a slight defect structure of both the precursor and the final carbon fiber and to a decrease in tensile strength to 0.88 GPa, but an increase in the elastic modulus to 110 GPa. Obviously, these characteristics are lower than for typical solution-spun carbon fibers [152]; however, firstly, the diameter of the fibers was significantly higher (>12 μm), and TOS and carbonization parameters can be optimized. In addition, the precursor fiber requires drawing to increase its orientation and crystallinity.

The other ways to solve the problem of TOS without melting the fiber can be divided into three groups: the use of external plasticizer; irradiation of spun fiber; and additional chemical modification of spun fiber.

### 4.2. Plasticization and Irradiation Approaches

#### 4.2.1. General Notes

The first includes plasticization of copolymers of AN with external plasticizer. Then, the removal of the plasticizer during drying or washing of the fiber results in the growth of the melting point above the TOS temperature. Thus, it becomes possible to carry out TOS and subsequently its carbonization without disturbing the shape of the fiber [97].

The second group includes methods based on the irradiation of fibers with different sources. As a result of irradiation, the radicals arise that can initiate cyclization or form crosslinks between macromolecules leading to the loss of their ability to flow. For example, it is known that in the course of radiation polymerization of AN in air at room temperature and a dose rate of 8.5 kGy/h (^60^C;o), crosslinking of the polymer occurs [153]. The authors discovered an unusual fact: it goes not only along the C–C bonds of the backbone due to the abstraction of a hydrogen atom from the CH group, but also along the C≡N groups that was confirmed by IR spectroscopy by the appearance of a band at 1670 cm^−1^ corresponding to vibrations of the –C = N–N = C– group formed by the recombination of two nitrile groups of neighboring macromolecules. Obviously, this approach can also be used for the finished fiber.

Irradiation with fast neutrons of HT-S carbon fibers from Courtaulds can lead both to an increase in the strength of fibers and to their oxidation and destruction depending on the atmosphere and temperature [154]. The tensile strength of the fibers increases upon irradiation dose in an atmosphere of liquid nitrogen at −196 °C or gaseous helium at 175 °C and decreases in air the more and lower the temperature. Irradiation in air turned out to be equivalent to heat treatment above 300–350 °C.

#### 4.2.2. UV Irradiation

UV irradiation of the copolymer acrylan in a nitrogen atmosphere or in a vacuum at wavelength of 253.7 nm leads to the release of a number of volatile products and the formation of a crosslinked polymer [155]. In this case, both the C–C bonds of the backbone are broken and the mobile hydrogen atom and even the nitrile group are split off (Figure 3).

The photo destruction of PAN of various molecular weights with M_n_ = 26–99 kg⋅mol^−1^ in solution of the mixture of ethylene and propylene carbonates in vacuum at wavelength of 253.7 nm (mercury lamp) was studied in [156]. For most cases, the irradiation of PAN led to the breaking of the backbone bonds and to decrease in the intrinsic viscosity of the polymer, however, in some cases, crosslinking of the polymer was observed, leading to an increase in viscosity. The photo-oxidative destruction of PAN leads to crosslinking of the polymer and the formation of a ladder structure by a mechanism similar to TOS, especially at moderate temperatures [157].

A detailed comparison of carbon fibers obtained from precursors prepared by solution and melt spinning, irradiated with UV light, and subjected to TOS and carbonization was described in [158]. In both cases, AN and MA copolymers were used, but with different compositions and MW. The commercial precursor obtained by “wet” spinning contains ~4 mol.% MA and has an intrinsic viscosity of 1.98 dl/g. The copolymer with the mole fraction of MA equal to 12% and an intrinsic viscosity of 0.49 dl/g is used for melt-spinning. Obviously, the mechanical characteristics of precursors and final carbon fibers will not coincide. Nevertheless, by selecting the conditions for UV irradiation and TOS for each of the polymers (precursors), similar trends are obtained. UV irradiation leads to partial degradation of the polymer, which is expressed in a decrease in tensile strength and elongation at break compared to the original precursor. It should be noted that crosslinking occurs unevenly throughout the fiber, as evidenced by the different ratio of the intensities of the absorption bands of the CH_2_ and CN groups during scanning along the cross section of the fiber. Tensile strength increases and elongation at break decreases after TOS. Finally, carbonization leads to an approximately three-fold increase in tensile strength and a decrease in elongation at break by one order compared to the original precursor. In general, the mechanical characteristics of the carbon fiber obtained from a copolymer of a higher MW and with a lower MA content (tensile strength 1 GPa, elastic modulus 190 GPa) turned out to be three times higher than that of a melt precursor with a lower MW and a high MA content (tensile strength 330 MPa, elastic modulus 60 GPa). Thus, UV irradiation helps to solve the problem of TOS for melted PAN.

A textile fiber manufactured by Taekwang Industry based on an AN–VAc copolymer (15 wt.%) was irradiated with UV source of different power at different temperatures below and above the glass transition temperature [159]. The irradiated fibers were subjected to stabilization and carbonization. As a result of irradiation, the time required for TOS is reduced to half an hour. The carbonized fiber had good mechanical characteristics: a tensile strength of 2.4 GPa and an elastic modulus of ~200 GPa.

Similar effects, i.e., the formation of crosslinks and partial cyclization, are also observed upon irradiation, which was demonstrated for a Mitsubishi fiber based on an AN copolymer and acrylamide [160].

#### 4.2.3. Electron Beam Irradiation

Electron beam irradiation is also used for pretreatment of PAN [161]. The effect of irradiation is reduced to the appearance of radicals in the polymer, which can be detected by the ESR method [162,163,164,165,166]. The ESR spectra contain complex multiplets, which some authors attribute to carbon-centered radicals, and others to a mixture of carbon-centered and imine (nitrogen-centered) radicals. Summarizing the literature data, the general scheme for the formation of radicals and their termination after electron beam irradiation can be represented as follows (Figure 4).

Thus, as a result of irradiation, crosslinking of chains and the formation of imine radicals that are able to initiate cyclization are possible. More recent studies have shown that polymer is heated during irradiation, which makes cyclization a very probable process [166]. Crosslinking and partial cyclization results in a shortened TOS time compared to a non-irradiated polymer. For example, the duration of TOS decreased by about third after electron beam irradiation of PAN and terpolymer of AN with MMA and itaconic acid [162]. In addition, irradiation leads to a decrease in the activation energy of cyclization, an increase in the degree of cyclization, and a decrease in fiber defects [167]. These effects increase with increasing dose rate, as was shown on the Jilin precursor (3000 filaments) that seems to be a copolymer of AN and MA according to IR spectroscopy and DSC data. The color of the fiber changes to slightly yellow after irradiation at room temperature at an irradiation dose of 200 kGy, and at temperatures of 180 and 200 °C, the color becomes already dark brown, while the non-irradiated fiber remains only slightly yellow. The irradiated fibers lose their solubility in DMF. The influence of the dose (100–250 kGy) on the degree of cyclization for this precursor was studied in more detail in [168]. As the dose increases to 150 kGy, the degree of cyclization increases to 0.27 and then reaches a plateau. In parallel with this, there is a change in both the surface of the fiber from smooth to grooved and tensile strength, which passes through a maximum at a dose of 200 kGy.

The reason for the TOS shift to low temperatures may come due to formation of free radicals upon irradiation, which have low mobility and are “imbedded” in the glassy polymer matrix, but their mobility appears when the temperature rises above the glass transition temperature [163]. In the presence of oxygen, the carbon-centered radicals are converted into oxygen-centered (peroxide) radicals and initiate cyclization at relatively low temperatures, resulting in the reduction of the duration of TOS. An increase in the irradiation dose leads to heating of the sample, and cyclization can occur during irradiation [165].

In most of the studies, PAN precursors from different manufacturers were used, including fairly cheap textile fibers. When choosing a precursor, the authors took into account the comparative analysis performed in [169]. Among three types of textile fibers based on the AN terpolymer, MA (4.7–6.0 mol.%), and the third comonomer (itaconic acid, 1.3 mol.%; sodium metallylsulfonate, 1 mol.%; sodium 2-methyl-2-acrylamidopropanesulfonate, 1 mole%), the best carbon fiber performance can be achieved with itaconic acid. For example, a commercial PAN precursor, which contains a small amount of itaconic acid according to IR spectroscopy and DSC data, was irradiated with a dose of 200 kGy followed by a short heat treatment at 230 °C for half an hour and carbonization in a nitrogen atmosphere at 1200 °C. Finally, carbon fiber with a tensile strength of 2.3. GPa and an elastic modulus of 216 GPa was obtained [163].

PAN precursor fibers (SAF 12K fibers, 12,000 filaments) based on the terpolymer of AN, MA (6 wt.%), and itaconic acid (1.2 wt.%) from the Courtaulds company was first irradiated by electron beam in air at room temperature, then it was passed through four-stage TOS at 210, 225, 245, and 263 °C and after that to low-temperature (400–700 °C) and carbonization in inert atmosphere (1250–1350 °C) [170]. The density of the irradiated fibers increases consistently at each of the four stages of TOS and becomes higher than for the non-irradiated samples. However, after carbonization, the mechanical characteristics of the irradiated fibers turned out to be 10–20% lower than those of the non-irradiated ones: the tensile strength was 2.85 GPa and the elastic modulus was 203 GPa.

Continuous electron beam irradiation (dose 1000 kGy), stabilization, and carbonization on a kilogram scale of textile fibers based on PAN was carried out in [171]. The authors managed to produce carbon fiber with a tensile strength of 3.1 GPa and an elastic modulus of 212 GPa, which turned out to be a good cheaper alternative to the T300 fiber from Toray (tensile strength 3.5 GPa, Young’s modulus 230 GPa). As textile fibers, three Dralon samples representing a terpolymer of AN, MA (3.8 mol.%), and sodium metallylsulfonate (0.2 mol.%), a copolymer of AN and Vac (4.1 mol.%), and PAN were spun by wet and dry–wet methods and subjected to irradiation. Irradiation facilitated the flow of TOS as noted above, resulting in an increase in tensile strength, elasticity modulus, and the formation of a more uniform carbon fiber structure.

A terpolymer of AN, MA (6 wt.%), and itaconic acid (1.2 wt.%) SAF (Courtaulds) subjected to electron beam treatment (dose 1200 kGy) and without it was used to produce carbon fibers [166]. Before stabilization, precursors differ in the amount of ketenimine C = C = N and conjugated C = N–N = C fragments, which can prevent cyclization. The mechanical characteristics of the non-irradiated carbon fiber were slightly higher than those of the irradiated one. For the latter, after optimization of TOS conditions tensile strength was equal to 2.27 GPa and elastic modulus to 174 GPa.

PAN precursor fibers from Anshan East Asia Carbon Fiber with a diameter of 13–21 µm were used in [172]. After electron beam irradiation of the fiber with a dose of 1000 kGy, it was subjected to TOS for 20 min at 200 °C and for 40 min at 250 °C, and then to carbonization in a nitrogen atmosphere for 1 h at 1200 °C. Finally, carbon fiber was produced with a tensile strength of 2.3 GPa. The same team studied the effect of additional tension on the PAN precursor fibers during irradiation [173]. As a result, crystalline structures developed during EBI stabilization. Tension led to a small increase in strength due to conversion of disordered structures into ordered structures in the fibers. 

Nanofiber mats with diameters of 400 nm obtained by PAN electrospinning (MW 150 kg⋅mol^−1^) from a solution in DMF were subjected to electron beam treatment with a dose of 500 to 5000 kGy and then carbonized for 1 h at 1000 °C [174]. The irradiated PAN nanofiber mats kept morphological behavior after the carbonization process.

Among the precursors studied, textile fibers based on an AN–VAc copolymer (15 mol.%) produced by Taekwang Industry should be noted [164]. These fibers have a melting point below TOS. However, the preliminary electron beam processing results in the partial cyclization of the fibers and, hence, to the loss of their melting ability. Furthermore, the radicals “captured” by the matrix upon irradiation provide both the rapid TOS at lower temperatures and higher conversions of nitrile groups during cyclization. Crosslinking in its turn leads to a reduction of the elasticity of the fiber prior to cyclization and significant shrinkage during the formation of the ladder structure. The carbon fiber obtained after carbonization is characterized by tensile strength equal to 1.8 GPa, an elastic modulus of 147 GPa, and an elongation at break of 1.3.

The entire cycle from polymer synthesis to carbon fiber production using melt spinning and electron beam irradiation is described in [62]. A copolymer of AN with MA (15 mol.%) with M_n_ ~30 kg⋅mol^−1^ and Đ > 2 was produced by emulsion polymerization. Its melt extrusion at 185 °C yielded a fiber of 15–20 µm in diameter (tensile strength 0.3 GPa, elastic modulus 6.8 GPa, and elongation at break 17.6%), which was then irradiated with a dose of 1500 kGy and subjected to stepwise TOS, heating the fiber from 160 to 250 °C. At the final stage, the stabilized fibers were carbonized in an inert atmosphere at 1200 °C. The carbon fiber had a homogeneous structure and a tensile strength of 1.37 GPa and an elastic modulus of 110 GPa.

#### 4.2.4. Plasma Discharge Treatment

Another alternative is the plasma discharge treatment [175,176]. In the plasma discharge of an argon–oxygen mixture, hydroxyl radicals, atomic oxygen, and ozone are generated, which makes it possible to expect the oxidation of a polymer exposed to plasma. The combination of TOS and plasma accelerates the stabilization process: in the absence of plasma treatment, the degree of cyclization after 15 min of the reaction at 230 °C is 0.28, and under the same conditions, however, with plasma treatment, it is 0.68 [176]. The same value of the cyclization degree is reached only within 2 h under conventional heating. The mechanical characteristics of carbon fibers produced after carbonization of stabilized precursors with and without the use of a plasma discharge differ. In the first case, the strength is 2.8 GPa and the elastic modulus is 204 GPa, while in the second case, the strength is 2.1 GPa and the elastic modulus is 196 GPa.

### 4.3. Additional Chemical Modification of Spun Fiber

The third group of methods for implementing TOS to melt-spun PAN without melting the fiber includes additional chemical modification of polymer. Aiming for this, a third comonomer with a photosensitive functional group is introduced at the stage of synthesis into the melt-processable AN copolymer with melting point below the onset of TOS [63,177,178,179,180]. The first information about the possibility of crosslinking of AN copolymers containing an embedded photosensitive component appeared in the early 2000s [72,181,182]. However, a detailed description of these processes followed later. As a comonomer capable of generating radicals upon irradiation, acryloylbenzophenone (Figure 6) is most often used in amounts not exceeding 1 mol.%.

It is worth highlighting the publication [183] in which a number of other polymerizable photo initiators (Figure 7) are proposed, such as organic carbonates and carbamates with vinyloxycarbonyl groups that can copolymerize with acrylic and methacrylic monomers, and whose main absorption maximum lies in the range of 250–280 nm.

The use of such monomers for melt PAN technology requires their thermal stability up to ~200°C, which corresponds to the melt spinning temperature. This aspect was studied in [177], where the authors compared a binary copolymer of AN with MA (15 mol.%) and two terpolymers of AN, MA (14 mol.%), and acryloylbenzophenone (1 mol.%) with M_n_ = 16–26 kg⋅mol^−1^ and Đ = 1.7–2.5. The melt stability is maintained in the temperature range of 200–220 °C with the introduction of the third monomer. However, the temperature at which the increase in viscosity due to TOS occurs is lower for terpolymers than for a copolymer with MA. This effect is enhanced with an increase in the MW of the terpolymer.

The second issue is the efficiency of UV irradiation after melt spinning. A copolymer of AN with MA (15 mol.%) and a terpolymer containing AN, MA, and 1 mol.% acryloylbenzophenone subjected to melt spinning at 210 °C are described in [178]. The terpolymer was irradiated at 110 °C with a low power (100 W) mercury lamp for 3 h or high power (4.5 kW) for 50 s. Then, the fibers based on the copolymer and the irradiated terpolymer were subjected to TOS and two-stage carbonization under similar conditions. The number of nitrile groups in the irradiated terpolymer decreases and unsaturated C = C bonds appear, which indicates the development of a ladder structure. A low power lamp ensures the crosslinking reaction, and a high-power lamp additionally provides the initiation of cyclization. High-power irradiation leads to the appearance of defects in the fibers, which persist even after carbonization and lead to a decrease in the mechanical characteristics of carbon fibers (tensile strength 300–450 MPa, elastic modulus 50–65 GPa, and elongation at break 0.5–0.9%).

An attempt was made to optimize this process by varying the MW of the terpolymer and the drawing ratio of the fiber during spinning [63]. As a result, carbon fibers were produced with ~7 nm in diameter, tensile strength of ~600 MPa, elastic modulus of ~130 GPa, and elongation at break of 0.4%.

The same approach was implemented to produce non-woven mats and to transform them into carbon mats [179]. Fibers were obtained by extrusion of a terpolymer of a similar composition, followed by their pressing, UV irradiation, stabilization, and carbonization. The mechanical characteristics of the carbon mats are as follows: linear density 32 tex/mm, tensile strength 1 cN/tex, elastic modulus 109 cN/tex, and elongation at break 1.3%.

Both the composition of terpolymer and the method of its synthesis were changed in [180]. Suspension polymerization in a mixture of DMSO and water was used instead of emulsion polymerization. Terpolymer AN with contents of MA 15 mol.% and acryloylbenzophenone of 1 mol.% was extruded at 200–230 °C followed by irradiation of the fiber at 110 °C with a 150 W mercury lamp for 2 h, then TOS at 310 °C and carbonization proceeded. The carbonized fiber with a diameter equal to 8 µm has tensile strength of 600 MPa, elastic modulus of 7.3 GPa, and elongation at break of 0.5%. 

In conclusion, it is worth citing the results of [184]. The authors used a mixture containing terpolymer AN, MA (14 mol.%), and acryloylbenzophenone (1 mol.%) and 1 wt.% of isotropic naphthalene pitch. Unfortunately, the authors limited themselves to studying the stages of spinning and irradiation, which does not allow one to assess the prospective for using pitch. However, a fiber with a diameter of 50 µm and a tensile strength of 240 MPa was obtained by spinning from the melt, and an increase in the pitch content leads to a decrease in strength and does not affect elongation at break. After UV irradiation of the fibers, the content of the gel fraction reaches 60% after 80 min and then decreases, which is apparently due to the partial degradation of the polymer. The influence of UV irradiation on the cyclization process in an inert atmosphere and in air is insignificant, and the strength of the fibers decreases with increasing irradiation time. Perhaps, the experimental conditions are not optimized. 

## 5. Conclusions

PAN precursors, their based carbon fibers, and composites are still attractive objects of investigation for many scientific groups. That is why numerous reviews have been published during the last decade; some of them are listed in the Introduction [2,19,20,21,22,26]. The vast majority of them summarize the wet-spun PAN fibers, while melt-spun PAN fibers are either ignored or hardly mentioned. The reviews that analyze the ways to fabricate low-cost carbon fibers are devoted mostly to various polyolefins or biobased polymers [3,4,5]. Furthermore, despite growing interest of the melt-spinnable AN copolymers and their application in carbon fiber production, we have not discovered any attempts to summarize recent achievements in this field except publication in the Journal of Textile Association [65]. However, its information became quite out-of-date. 

Thus, in this review, we have considered various approaches to the production of melt-spun AN copolymers and their based carbon fibers and summarized the recent patent information. 

To obtain melt-spun AN copolymer, one needs to decrease the melting point of the polymer below its cyclization temperature. This task can be solved easily by internal or external plasticization. A wide range of vinyl monomers are used as internal plasticizers including vinyl acetate, methyl acrylate, etc. Notably, their fraction in copolymer is several-fold greater than in traditional wet-spun AN copolymers. Regarding external plasticizers, the necessity of their removal from the fiber by washing reduces the advantages of melt-spinning to zero due to appearance of the stages of recovering and purification of plasticizer and the solvent (water).

If melt-spun fibers are used as precursors for carbon fibers, their stabilization should proceed without melting. This new requirement forces researchers to search for other solutions. Among them, several approaches can be distinguished. The first way is the addition of acidic comonomer that provides rapid TOS. The second is the rise of the melting point through removal of the external plasticizer. Finally, crosslinking of the spun fiber by irradiation of various nature may be used. The latter seems more promising, even if it requires additional equipment for irradiating the fibers.

It is surprising that melt-spinning of AN copolymers has not been implemented yet into full-scale production of carbon fibers. Based on the analysis of the scientific and patent (see below) literature, we believe that it should happen for the foreseeable future.

## Data Availability

The data presented in this study are available on request from the corresponding authors.

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
