# Peer review of "Melt-Spinnable Polyacrylonitrile—An Alternative Carbon Fiber Precursor"

_polymers, 2022, doi:10.3390/polym14235222_

Round 1
Reviewer 1 Report
This review paper discusses melt-spun PAN fibers as precursor of carbon fibers. The author summarized chemical and physical approaches to reduce melting point of PAN-based copolymers to make them processable, and discussed how to address thermal oxidative stabilization. This article is well organized. The only issue is the numbering of conclusion should be 5 instead of 6.
Author Response
Dear reviewer,
Thank you for your comment. We have changed numbering according to your note.
Reviewer 2 Report
This manuscript can be accepted.
Author Response
Dear reviewer,
Thank you for your recommendation.
Reviewer 3 Report
The review work titled “Melt-spinnable Polyacrylonitrile – an Alternative Carbon Fiber Precursor”, is really interesting, but there are some needs to be added before accepting this review work.
1. The abstract must incorporate some key points about the review work. I mean overall aspects of the review work must be included.
2. Introduction part needs to be incorporated with carbon fibers for polymer composite applications. See the below works and follow that
https://doi.org/10.3390/polym14061095
DOI: 10.1002/pc.27150
3. There must be proper explanation for each titles mentioned. In title 4 nothing was described.
4. I feel subtopics in the work is really lesser. Add few more sub divisions to make the work better.
5. Is that relevant to use these much of article review. Some works are not used properly.
6. Conclusion must include overall analysis of all the reviews.

Author Response
Dear reviewer,
We are thankful for your fruitful comments and have modified the paper in accordance with your recommendations. All the changes in the text are marked with red.
- We have extended the Abstract.
- We have added corresponding sentence to the Introduction and added the new reference. As a result, we have changed the numbering of all the references throughout the text and in the list of references.
- We have added the proper explanation where necessary and modified some titles.
- We have added some subdivisions in sections 3 and 4.
- We have carefully used all the references aiming to give clear idea about state-of-art.
- We have changed the conclusions and analyzed additionally the reviews cited in introduction.